# Comparing FY-2F/CTA products to ground-based manual total cloud cover observations in Xinjiang under complex underlying surfaces and different weather conditions

Shuai Li[1], Hua Zhang[2*], Yonghang Chen[1], Zhili Wang[2, 3], Xiangyu Li[4], Yuan Li[5], Yuanyuan Xue[5]

[1]College of Environmental Science and Engineering, Donghua University, Shanghai 201620, China

[2]State Key Laboratory of Severe Weather, Chinese Academy of Meteorological Sciences, Beijing 100081, China

[3]Key Laboratory of Atmospheric Chemistry of CMA, Chinese Academy of Meteorological Sciences, Beijing 100081, China

[4] School of Materials and Chemical Engineering, Pingxiang University, Pingxiang, Jiangxi 337000, China

[5]China energy Xinjiang Jilintai Hydropower Development Co., Ltd, Nilka, Xinjiang 835716, China

*Correspondence to:* Hua Zhang (huazhang@cma.gov.cn)

**Abstract.** Clouds are an important parameter of artificial water augmentation, which are of substantial significance to judge the precipitation capacity. Xinjiang is an arid region in Northwest China, where weather stations are sparsely distributed, the types of underlying surface are complex and the climate difference between southern and northern Xinjiang varies greatly. However, the retrieval of the total cloud cover (TCC) from satellite in arid areas is a challenging task. Based on the data of TCC observed by ground observation stations (GOS) from June 2015 to May 2016, considered the complex underlying surfaces and different weather conditions, the precision, consistency and error between the cloud total amount of FengYun-2F stationary satellite (FY-2F/CTA) and manually observed TCC are compared and evaluated in Xinjiang region. The findings of this study are as follows: (1) The precision rate (PR) of FY-2F/CTA in Xinjiang region is 75.6 %, which gradually decreases from north to south, demonstrating a high false rate (FR) and a low missing rate (MR); The consistency rate (CR) is 51.5 %, with little difference among three sub-regions of Xinjiang, all showing high weak rate (WR) and low strong rate (SR), which means that the TCC inverted from FY-2 satellite data are generally lower than those observed by GOS, especially in Southern Xinjiang; The Bias is -20 %, and all the error indexes (EIs) including Bias, MAE and RMSE increase from central to the north and south of Xinjiang, that means the EIs are the lowest in Tianshan Mountains, and the highest in Southern Xinjiang. FY-2F/CTA exhibit higher PR and CR in the underlying surface of vegetation compared to non-vegetation, that is to say, FY-2F/CTA perform best in the underlying surfaces of forest and plowland, while their performance are relatively poorer in the underlying surface of snow and ice. (2) With rising temperature the PR and CR of FY-2F/CTA increase, while the EIs decrease; Under various temperature conditions, FY-2F/CTA have always been exhibiting high MR, low FR (on the contrary in January), high WR and low SR. From low elevation to high elevation, the PR and CR of FY-2F/CTA decrease, but the PR increases significantly when the altitude is higher than 2000 m. (3) Dust reduces the CR of FY-2F/CTA, increases their WR and MR, but has a relatively minor impact on the identification of cloud and non-cloud. (4) Under different cloud cover levels, the PR and EIs of FY-2F/CTA are directly proportional to the amount of TCC, while the CR is inversely proportional to it, that is, the CR is higher, the PR and EIs are lower under clear sky and partly cloudy conditions, and the CR is lower, the PR and EIs are higher under cloudy and overcast conditions. This study assessed the

FY-2F/CTA under various conditions in arid areas of Xinjiang, including complex underlying surface, various temperature and altitude, dust effects and different cloud cover levels. Thus, the research finding would serve as a valuable reference for satellite-based retrieval and applications related to TCC in arid regions.

## 1 Introduction

Clouds are one of crucial element of weather and climate characteristics, they affect atmospheric movement and the earth's climate through three mechanisms: radiative forcing (Li et al., 2019; Kiran et al., 2015; Haynes et al., 2013), latent heat forcing (Loeb et al., 2018) and convective forcing (Slingo and Slingo, 1988; Guo et al., 2015; Li et al., 2017). In addition, clouds can influence the climate system indirectly through interactions with aerosols (Boers et al, 2006; Zhou et al, 2020), the feedback effect of climate change can also directly lead to change in cloud characteristics simultaneously (Harrison et al., 1990; Errico et al., 2007; Chepfer et al., 2014), for every 1 °C increase in global temperature, the content of water vapor in the atmosphere will increase by about 7 % (Boucher et al., 2013), the variation of water vapor can affect the occurrence, development and extinction of clouds, and change cloud cover, cloud albedo and cloud microphysical properties (Zhang et al., 2022). The World Climate Research Programme (WCRP) lists understanding clouds, atmospheric circulation and climate sensitivity as one of the key scientific challenges currently facing the climate community (Bony et al., 2015), as an important component of climate feedbacks, cloud feedbacks are one of the largest sources of uncertainty in simulating current climate and predicting future climate change (Bony et al., 2006; Zelinka et al., 2017). Cloud cover, as one of significant cloud parameters, can directly or indirectly affect the research on climate change, weather or climate models by taking advantage of the cloud macro and micro parameters, radiation parameters, water vapour and precipitation, aerosols and other physical quantities (Chen et al., 2000; Betts et al., 2014; Ceppi et al., 2017). An accurate understanding of the spatial distribution and temporal evolution of cloud cover is the basis for conducting research on atmospheric radiation, energy and water cycles, climate analysis and numerical models. It is of great significance for gaining insight into the complex interactions between clouds and the terrestrial system, and thus for better understanding climate change (Pavolonisand Key, 2003; Ronald et al., 2007; Guo et al., 2015b).

At present, the conventional ground-based cloud cover observations, global reanalysis cloud products (NECP, ERA, etc.) and the total cloud cover (TCC) retrieved by satellite remote sensing (including ISCCP, MODIS, NOAA series and CloudSat, etc.) are the most common cloud cover data. They are extensively applied in analysis of cloud parameters and climatological characteristics (Sun, 2003; Ding et al., 2004; Danso et al., 2019), satellite inversion and validation (Yousef et al., 2020; Yang et al., 2020), in particular, the cloud parameters of satellite inversion are usually compared with those of cloud radar, lidar, spectral imager and ground observations (Kotarba, 2009; Wang and Zhao, 2017). A great deal of research has been carried out by experts in the field of inspection and evaluation of cloud products. Werkmeister et al. (2015) compared cloud

fractional cover measured by radiometers on polar satellites AVHRR and on one geostationary satellite SEVIRI to ground-based manual SYNOP and automated observations by a cloud camera Hemispherical Sky Imager, and found in general good agreement between satellite-derived estimated compared to Hemispherical Sky Imager with biases ranging from 2 % (AVHRR) to 8 % (SEVIRI) and standard deviations of 22 % (AVHRR) and 29 % (SEVIRI) for instantaneous results, and also found that SYNOP underestimated cloud fractional cover by 6±19 % compared to Hemispherical Sky Imager and SEVIRI. Free et al. (2016) compared a homogeneity-adjusted dataset of TCC from weather stations in the contiguous United States with cloud cover in four global reanalysis products, including the Climate Forecast System Reanalysis from NCEP (CFSR), the Modern-Era Retrospective Analysis for Research and Applications from NASA (MERRA), ERA-Interim from ECMWF (ERA-Interim), and the Japanese 55-year Reanalysis Project from the Japan Meteorological Agency (JAR-55), the result showed that TCC from ERA-Interim and CFSR had the best correlation with weather station, followed by JRA-55 and MERRA had the lowest correlation. Sun et al. (2015) concluded that average cloud cover in the United States from AVHRR Pathfinder Atmospheres Extended (PATMOS-x) with station observations, including 54 National Weather Service (NWS), Federal Aviation Administration (FAA) stations and 101 military stations, showed the closest trend, followed by ISCCPs, with CLARA-A1 (AVHRR Data Edition 1) showing a larger negative trend. Wu et al. (2014) found that in the United States, Active Remote Sensing of Clouds (ARSCL) and International Satellite Cloud Climatology Project (ISCCP) have higher cloudiness than surface observations, this was the same as the research results in China, where the ISCCPD2 total cloud cover products (TCCPs) were 8.46 % higher than the ground observation (Wang and Wang, 2009), the accuracy of ISCCP TCCPs in January is better than that of MODIS, while the MODIS TCCPs in July was slightly better than that of ISCCP (Liu et al, 2009), NOAA/AVHRR TCCPs could better reflect the variation characteristics of TCC in China, However, the amount of TCC was slightly lower than that observed at the ground observations (Liu et al., 2016). In addition, many experts have done a lot of research in the development and validation of TCCPs of FY-2 series satellite, and some research results indicate that in China, FY-2 satellite observed cloud products were lower relative to ground-based manual observations and also slightly lower than the MODIS calculated TCCPs, but the overall trend was comparable (Li et al., 2018; Liu et al., 2017; Han et al., 2015). However, the results of Xi et al. (2013) showed the data set of TCCPs in East Asia retrieved by FY-2 and GMS were higher in magnitude than that observed by the ground-based observations in the north of 40°N. Few studies related to clouds in the Xinjiang region, and only some scholars have analyzed the cloud characteristics of Xinjiang, Feng et al. (2014) analyzed the characteristics of different types of cloud heights in Xinjiang mountainous area by using CloudSat data; Based on FY-2F data, Li et al. (2019) analyzed the distribution and difference of cloud cover in mountainous and basins of Xinjiang.

Xinjiang is a typical arid and semi-arid region, and the shortage of water resources has become a "bottleneck" problem in the development of social economy and ecological construction in Xinjiang. It is a vast and sparsely populated area with extremely low spatial coverage rate of ground-based conventional observation stations, which is more suitable for satellite

observation. How to use satellite observations to complement ground-based observation has become an urgent issue. The accuracy assessment of cloud cover retrieved by satellite is the basis of application and also a challenging task. In this paper, the examination and evaluation of the cloud total amount of FengYun-2F stationary satellite (FY-2F/CTA) are carried out using ground-based manually observed TCC and considering complex underlying surface (subsurface types, temperature and

altitude conditions) and different weather conditions (dust effects and different cloud cover levels) with Xinjiang as the examination region, with a view to providing valuable references for the application and research of FY-2 cloud products.

## 2. Research area, data and methods

### 2.1 Research area

Xinjiang (Figure 1) is located in the hinterland of the Eurasia and characterized by a typical temperate continental arid

climate. It is a core component of the world's largest non-zonal arid zone (the Central Asian arid zone) and occupies an extremely important position in the northern hemisphere climate system, with an average annual precipitation of only about 150 mm, which is representative of the global arid zone (Yao et al., 2018). With a complex topography of "three mountains sandwiched by two basins" and a unique landscape, mountains and basins account for 42.7 % and 57.3 % of the total area of Xinjiang region, respectively. The annual average TCC in Xinjiang is 37.7 %, showing a pattern of "less in the east and more

in the west; less in the south, more in the north; less in plains and basins, more in mountains "(Li et al., 2019).

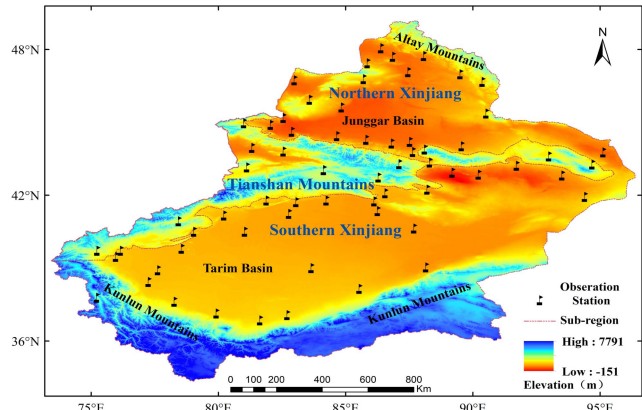

**Figure 1.** Overview of the geographic location and topography of Xinjiang with three mountain ranges (Altai Mountains, Tianshan Mountains and Kunlun Mountains) and two basins (Tarim Basin and Junggar Basin). The red dashed polygons indicate the three sub-regions (NX, SX, and Tianshan), Black flags represent the 66 TCC ground observation stations in the Xinjiang region.

### 2.2 Research data

The data used in this paper mainly include: the data of manually observed TCC by 66 ground observation stations (GOS) in Xinjiang from June 1, 2015 to May 30, 2016 obtained from the information center of Xinjiang, of these, 24 GOS are

distributed in Northern Xinjiang (NX), 10 in Tianshan Mountains (Tianshan) and 32 in Southern Xinjiang (SX). Manually observed TCC are the human-eye observations on the ground, they are collected five times a day (at 00:00, 03:00, 06:00, 09:00, and 12:00 UTC), with values ranging from 0 to 10. The observations follow the specifications outlined below: when the sky is entirely clear, the TCC is recorded as 0; if the sky is completely covered by clouds, it is recorded as 10; if the sky is fully covered by clouds but openings in the clouds allow glimpses of the sky, it is recorded as 10-; When there are a few clouds in the sky, amounting to less than 0.5 of the sky's coverage, the TCC is recorded as 0; When visibility is impaired due to phenomena such as haze, suspended dust, sandstorms, or blowing sand, rendering the determination of TCC either entirely or partially indiscernible, the TCC is recorded as " – ". If clouds occupy one-tenth of the sky, the TCC is recorded as 1; if they occupy two-tenths of the sky, it is recorded as 2, and so forth, following a similar progression for different levels of cloud coverage.

FY-2F/CTA customized from the data service network of National Satellite Meteorological Center (http://satellite.nsmc.org.cn/portalsite/default.aspx) from June 1, 2015 to May 30, 2016, the total number of data is 8317. Among them, FengYun-2F (FY-2F) is the fourth geostationary satellite developed by China independently. It is equipped with various detection channels, including visible light (0.5 - 0.9 μm), mid-wave infrared (3.5 - 4.0 μm), thermal infrared (infrared channel 1(10.3 - 11.3 μm), infrared channel 2(11.5 - 12.5 μm) and water vapor (6.3 - 7.6 μm)). The satellite provides observational data every half hour, allowing for improved monitoring of the entire process of cloud formation, development, and dissipation. The cloud products of FY-2F include cloud cover, cloud type, cloud top temperature, among others. FY-2F/CTA represents its TCC product, the spatial resolution is 0.1º × 0.1º, temporal resolution is 1 hour, and the projection method is equal latitude and longitude projection. This configuration enables enhanced monitoring capabilities for the complete lifecycle of clouds.

## 2.3 Research methods

FY-2F/CTA is calculated by formula Eq. (1). That is, firstly, by calculating the radiation value of clear sky and full cloud for a certain pixel; secondly, by using the radiation formula according to the actual radiation value of the pixel at a certain time (Liu et al., 2017).

$$TCC = (I - I_{clr})/(I - I_{cld}) \qquad (1)$$

Where TCC is the total cloud cover; $I_{clr}$ is the radiation value of full cloud pixels; $I_{cld}$ is the radiation value of clear sky pixels, I is the actual radiation value of the pixel at a certain time. This calculation method starts from the radiative transfer equation, takes into account the emissivity of the cloud, preserves the spatial resolution of the original observation image, and improves the subimage problem in principle, but the TCC computed by satellite will appear to be low when compared with the TCC observed by GOS.

The specific data processing methods are as follow. ① To mitigate the impact of short-term weather changes on ground observations, reduce data fluctuations caused by observational errors, and enhance data stability, the stations with continuous observations for 20 days or more are selected to enhance data stability (Liu et al., 2016); the abnormal observations, including missing data and outliers (observations < 0 or > 10 or the records are " – "), are removed from the dataset during the preliminary quality control of the ground observation data. ② The ground observation TCC reflects the cloud cover within a certain range around each observation point, the area can reach several kilometers or even more than ten kilometers, for satellite observation TCCPs, only the radiation ratio at grid points is considered. Therefor for satellite products, the TCC at each station is determined by averaging the cloud amounts of all grid points within a 10 km radius centered on the station's location. ③ Using the observation time, latitude and longitude information of the observation stations, the TCC observed by the GOS are matched with those observed by the satellite, and the total number of matched data is 80,855.

In addition, because FY-2F/CTA observations are provided as integer values from 0 % to 100 %, they are converted into tenths from 0 to 10, as listed in Table 1 (Kim et al., 2023).

**Table 1.** Tenth cloud cover conversion table of satellite (%) and ceilometer (okta) cloud cover.

| % | ≤5 | 5-15 | 15-25 | 25-35 | 35-45 | 45-55 | 55-65 | 65-75 | 75-85 | 85-95 | >95 |
|---|---|---|---|---|---|---|---|---|---|---|---|
| Okta | 0 | 1 | 2 | 2 | 3 | 4 | 5 | 6 | 6 | 7 | 8 |
| Tenth | 0 | 1 | 2 | 3 | 4 | 5 | 6 | 7 | 8 | 9 | 10- / 10 |

The matching principles are as follows. ① When the observation of ground station is clear sky and the satellite observation is also clear sky, it would be judged as an effective clear sky detection point of satellite, which is recorded as Nn. ② When the observation of ground station is cloud and the satellite observation is also cloud, it would be deemed as an effective cloud detection point of satellite, which is recorded as Yy. ③ When the observation of ground station is clear sky but the satellite detection result is cloud, it would be judged that the satellite misjudgment and be recorded as Ny. ④ When the observation of ground station is cloud, but the satellite detection result is clear sky, then the effective cloud arithmetic average is performed on the points in a certain area around the point, and if it is still resulted clear sky, then the satellite is judged to have missed the detection, this point is recorded as Yn.

The precision analysis of FY-2F/CTA include precision rate (PR), False rate (FR) and missing rate (MR), which are calculated by Eq. (2) to Eq. (4) respectively:

$$PR = \frac{Nn+Yy}{Nn+Yy+Ny+Yn} \times 100\% \tag{2}$$

$$FR = \frac{Ny}{Nn+Yy+Ny+Yn} \times 100\% \tag{3}$$

$$MR = \frac{Yn}{Nn+Yy+Ny+Yn} \times 100\% \qquad (4)$$

For the consistency analysis of FY-2F/CTA, if the absolute values of the difference between FY-2F/CTA and ground-based manual TCC observations are less than or equal to 2, they are considered to be correct; if the values of difference are greater than 2 or less than -2, they are considered to be stronger or weaker respectively (Li et al., 2018); Then the consistency rate (CR), strong rate (SR) and weak rate (WR) can be expressed as Eq. (5) to Eq. (7) respectively (Han et al., 2015):

$$CR = \frac{NR_k}{NR_k+NS_k+NW_k} \times 100\% \qquad (5)$$

$$SR = \frac{NS_k}{NR_k+NS_k+NW_k} \times 100\% \qquad (6)$$

$$WR = \frac{NW_k}{NR_k+NS_k+NW_k} \times 100\% \qquad (7)$$

Where $NR_k$ represents the number of times that FY-2F/CTA are correct for a station during the test period, $NS_k$ means the number of times that FY-2F/CTA are stronger and $NW_k$ means the number of times that FY-2F/CTA are weaker.

Using the ground-based manual TCC observations as true values, the error analysis between FY-2F/CTA and ground-based manual TCC observations include Bias, mean absolute error (MAE) and root mean square error (RMSE), which are calculated by Eq. (8) to Eq. (10) respectively:

$$Bias = \frac{1}{N}\sum_{i=1}^{N} (X - X_0) \qquad (8)$$

$$MAE = \frac{1}{N}\sum_{i=1}^{N} abs(X - X_0) \qquad (9)$$

$$RMSE = \sqrt{\frac{1}{N}\sum_{i=1}^{N} (X - X_0)^2} \qquad (10)$$

Where $N$ represents the number of matched samples, $X$ are the FY-2F/CTA observations and $X_0$ are the ground-based manual TCC observations.

## 3 Result and discussion

### 3.1 The difference between FY-2F/CTA products and Manual observed TCC in complicated underlying surface of Xinjiang

#### 3.1.1 Different underlying surface types

Due to the fragmentation of complex terrain, the climate in different regions of Xinjiang varies widely, therefore, this paper calculates and analyzes the precision, consistency and error of FY-2F/CTA in NX, Tianshan and SX (Figure 2). It can be seen that the PR of FY-2F/CTA in Xinjiang region is about 75.6 %, and that in NX, Tianshan and SX is 79.5 %, 75.5 % and 72.6 % respectively, showing a decreasing trend from north to south. The FR is higher than the MR overall, with the highest FR in the Tianshan and the highest MR in the SX. This conclusion is consistent with MODIS TCCPs, in contrast to the NOAA/AVHRR TCCPs. The study shows (Liu et al., 2017) that MODIS/Aqua TCCPs present high FR and low MR, mainly

due to the 1.375 μm channel of MODIS, which is dedicated to detecting cirrus cloud in the upper atmosphere. In addition, several channels such as 6.715, 7.325 and 13.935 μm, can be used to assist in the detection of cirrus cloud, reducing the MR of cirrus cloud detecting. However, the TCCPs that calculated by NOAA/AVHRR show high MR, mainly due to the translucent characteristics of thin cirrus cloud, whose reflectance of visible channels and bright temperature of infrared channels are not obvious. Of the five detection channels available to NOAA/AVHRR, although it is possible to detect thin cirrus cloud with the bright temperature difference of the infrared split window, the detection accuracy is not high.

The CR of FY-2F/CTA does not vary significantly among three sub-regions of Xinjiang, maintaining at about 51.5 %. The TCCPs observed by FY-2 satellite are lower than that observed by the GOS on the whole, that is, FY-2F/CTA products show high WR and low SR, and the WR is highest in SX, and the SR is highest in Tianshan. FY-2F/CTA products are calculated in a certain field of view by using the radiation value, which takes into account the emissivity of cloud and solves the problem of some sub-pixel cloud, but the effective cloud cover obtained from this calculation is relatively small; The ability of satellite to detect high cloud is better than that of low cloud, the main reason is that low cloud is closer to the ground and have less variability, the result of satellite cloud detection algorithm could easily misjudge low cloud as surface, which would underestimate the cloudage, while the Tarim Basin, which occupies most of SX, is dominated by stratocumulus (Li et al., 2019), so WR is much higher than SR in SX. This conclusion is the same as the NOAA/AVHRR TCCPs, and contrary to the MODIS/Aqua TCCPs. Compared with the TCC observed by the GOS, TCCPs calculated by NOAA/AVHRR are smaller, but TCCPs obtained by MODIS/Aqua present larger, and the TCCPs retrieved by FY-2 are lower than that retrieved by MODIS/Aqua. This may be due to the fact that NOAA/AVHRR TCCPs and FY-2/CAT use the same calculation method, and that the resolution and channel information of the two satellites are close. Meanwhile, the differences in observational capabilities and cloud detection algorithms between MODIS/Aqua TCCPs and the two satellite TCCPs mentioned above are the main reasons for the deviations (Liu et al., 2016, 2017).

The errors of FY-2F/CTA in three sub-regions of Xinjiang are high overall, and all the error index (EIs) including Bias, MAE and RMSE show increasing trend from the central to the north and south of Xinjiang, that is, the EIs are the lowest in Tianshan, and the correlation coefficient between FY-2F/CTA and ground-based manual TCC observations is high, about 0.65; While the EIs in NX and SX are high, and the correlation coefficients are slightly low, especially in NX, only 0.52 (Figure 3). This may be mainly due to the fact that satellite cloud observation is to observe from upper air downwards, and what is observed is mostly high cloud, while ground-based observation is to observe cloud from the ground upwards, and what is observed is mostly medium and low cloud. In addition, there is a certain subjectivity in the ground-based manual cloud observation, and when the cloud in the sky cannot be identified or not fully identified due to snow, fog, sand and other weather phenomena, they would affect the accuracy of cloud observation, make the difference between satellite observation and ground-based manual TCC observation larger. In this study, the reasons for the differences will be analysed in detail in terms of complex subsurface types, various temperature and altitude conditions, dust effects, and different cloud levels.

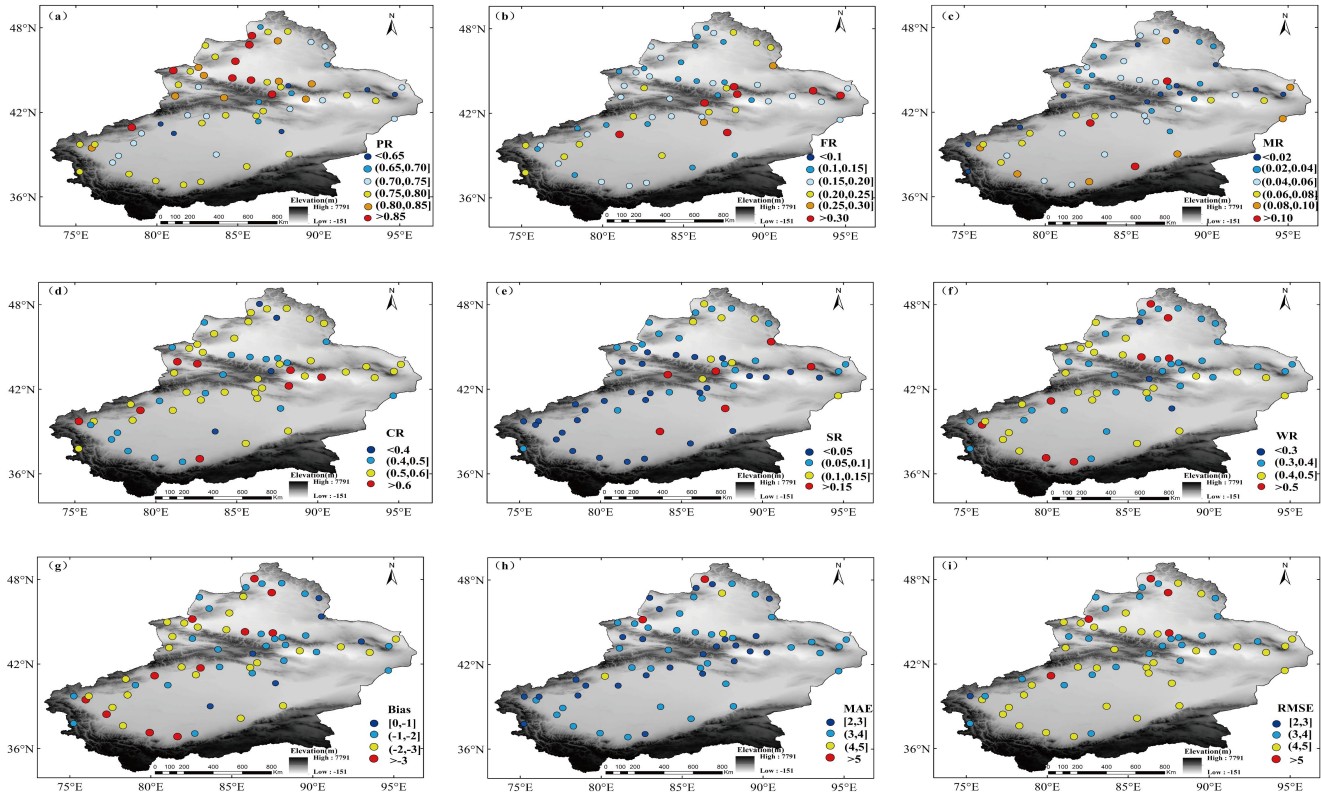

**Figure 2.** The precision, consistency and error spatial distribution map of FY-2F/CTA products in Xinjiang. Where, from Figures (a) to (i) denote PR, FR, MR, CR, SR, WR, Bias, MAE, RMSE respectively. The data is based on the hourly TCC of FY-2F/CTA and ground-based manual observations. The total number of all valid matches is 80855, among them, 29750 are distributed in NX, 10884 are distributed in Tianshan and 40221 are distributed in SX.

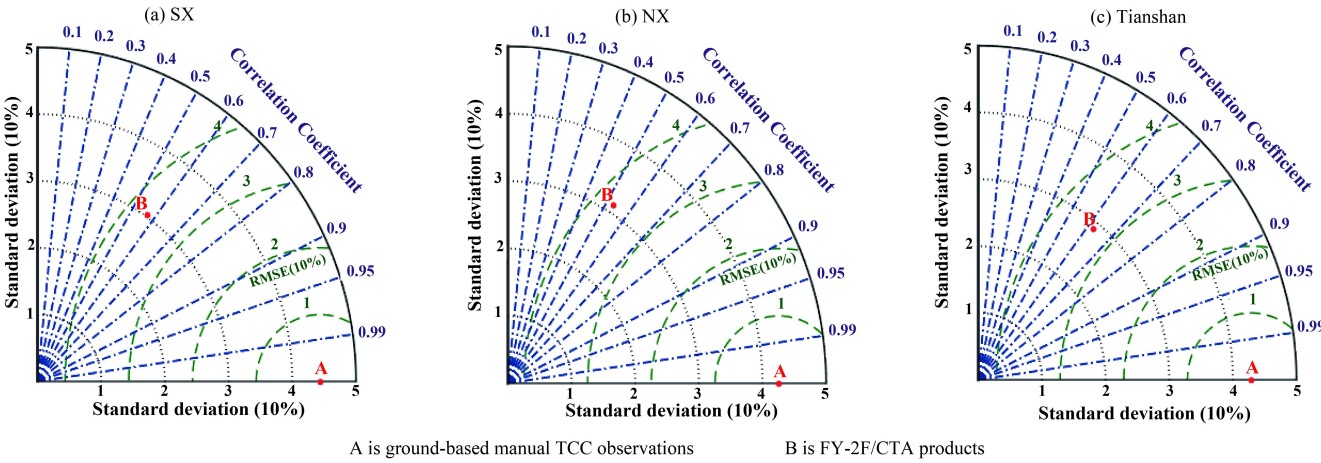

A is ground-based manual TCC observations    B is FY-2F/CTA products

**Figure 3.** Taylor diagrams between FY-2F/CTA products and ground-based manual TCC observations in different regions of Xinjiang.

Xinjiang is a vast territory with complex underlying surface types, in order to investigate the influence of diverse underlying surfaces on the difference between FY-2F/CTA and ground-based manual TCC observations, this study selects typical underlying surfaces in Xinjiang, including snow and ice, desert, city in non-vegetated region, grassland, forest and plowland

in vegetated region, compares and analyzes the precision, consistency and error of FY-2F/CTA under complex underlying surfaces (Figure 4). It can be seen that the PR, CR and EIs of vegetation region are better than those of non-vegetation region, with the highest accuracy in the underlying surface of forest and plowland, and the lowest accuracy in the underlying surface of snow and ice; The MR in the vegetation area is higher (grassland is the highest), and the FR in the non-vegetation area is higher (snow and ice is the highest); The WR is higher than the SR in all kinds of underlying surfaces, but the difference is not significant in the underlying surface of snow and ice. This is mainly due to the fact that the approximate distribution of albedo for different underlying surfaces range from 0.2-0.46 for desert, 0.15-0.25 for grassland, 0.1-0.2 for forest, 0.75-0.95 for snow (clean), 0.25-0.75 for snow (wet and dirty). And according to the extracted temperature results of Kang et al. (2022) using MOD11C3, the annual mean values of land surface temperature (LST) for different underlying surfaces are 2.57°C for grassland, 9.03°C for forest, 10.27°C for plowland, 11°C for city and -7.01°C for snow and ice respectively. In the process of FY-2 TCCPs retrieval, the underlying surface of snow and ice exhibits a high reflectance and low surface temperature, and the minimal contrast between the two factors, makes distinguishing between ice/snow and clouds challenging, particularly during nighttime and when visible light channel data is unavailable. Therefore, the PR of FY-2F/CTA in snow and ice underlying surface is low, and the FR is high. In contrast, the underlying surface of forest, which differs significantly from the underlying surface of ice and snow, tends to yield more effective cloud discrimination.

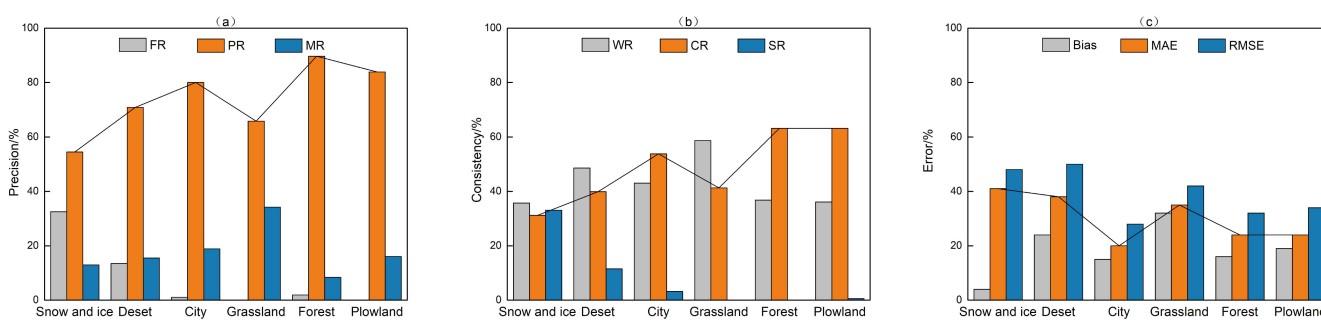

**Figure 4.** The precision, consistency and error of FY-2F/CTA products in complicated underlying surface of Xinjiang. In this case, the number of samples is 9196, of which 1650 are distributed in snow and ice underlying, 1596 in desert underlying, 992 in city underlying, 1653 in grassland underlying, 1653 in forest underlying, 1652 in plowland underlying.

### 3.1.2 Under different temperature conditions

Xinjiang belongs to temperate continental climate and plateau mountain climate with average temperatures of -4.46°C in January, 26.73°C in April, 38.66°C in July and 19.52°C in October (Kadiayi Alemu, 2021). In this paper the precision, consistency and error of FY-2F/CTA under different temperature conditions are analyzed by using January, April, July and October to represent winter, spring, summer and autumn respectively. Figure 5 is the scattered point of the FY-2F /CTA and ground-based manual TCC observations in Xinjiang region. It can be seen that the amount of TCC inverted from FY-2 satellite data are generally lower than that observed by GOS, with the greatest degree of underestimation in April; the

correlation between the two is the best in July and October, and is the worst in January, and all of them pass the significance test of 0.01 except for January. This is basically consistent with the test results of the North China and Huanghuai regions carried out by Han et al. (2015), that is, the correlation coefficient between FY-2/CTA and ground-based manual TCC observation is the largest in May at 0.92 and the smallest in January at 0.56.

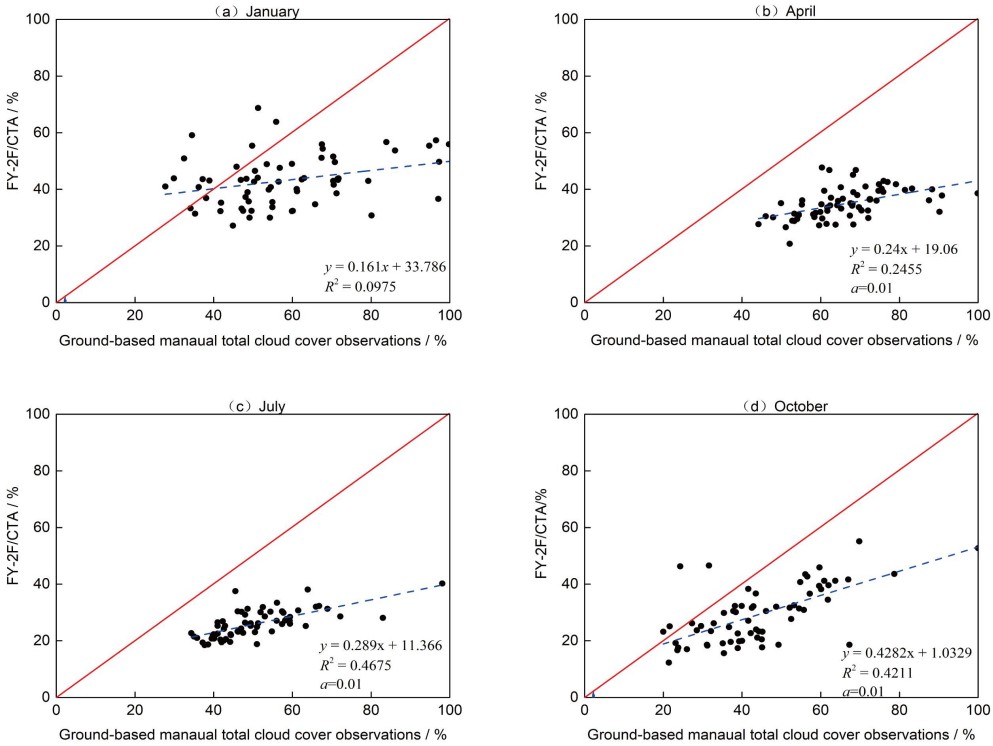

**Figure 5.** The scatter plot of FY-2F/CTA compared to ground-based manual TCC observations in Xinjiang. The data is the monthly average TCC of FY-2F/CTA and ground-based manual observations. It is based on the hourly data of FY-2F/CTA and GOS in January, April, July and October (the total sample points of 66 GOS are 7634, 7235, 7592, 7554, respectively), and after summing and average calculation, the monthly average TCC of FY-2F/CTA and ground-based manual observations of stations are obtained. Therefore, in this figure, the total number of all valid matches is 264, among them, 66 in January, 66 in April, 66 in July, and 66 in October.

Figure 6 shows boxplot of the precision, consistency and error of FY-2F/CTA under different temperature, it can be seen that the PR of FY-2F/CTA in Xinjiang region increases with the increases of temperature, for instance, in July, it reaches around 76 %, whereas in January, it is lower at 66 %; the PR tends to decrease from the central region towards the northern and southern regions; when temperature is above 0°C, it exhibits a high MR and a low FR, conversely, when temperature drops below 0°C, the pattern is reversed, that is, there's a high FR and a low MR in January. The CR is higher in July and October (52 % and 67 %, respectively) than that in January and April (50 % and 45 %, respectively), and it decreases from the central to the north and south of Xinjiang in April, increases from the north to the south of Xinjiang in July; Under various temperature conditions, it shows a high WR and a low SR. The EIs are high in January and low in July, which increase from the central to the north and south of Xinjiang, that is, the EIs are the lowest in Tianshan and highest in SX, and they are high

in January in three sub-regions of Xinjiang. This mainly caused by the fact that at higher temperature, the larger temperature

differences between underlying surface and cloud-top there are, the better the satellite discrimination there would be.

Additionally, in January, the region north of Tianshan and some mountainous areas in SX are covered with snow, therefor,

the PR of FY-2F/CTA in Xinjiang in January is low and its EIs are large, possibly due to the misjudgment of cloud and snow.

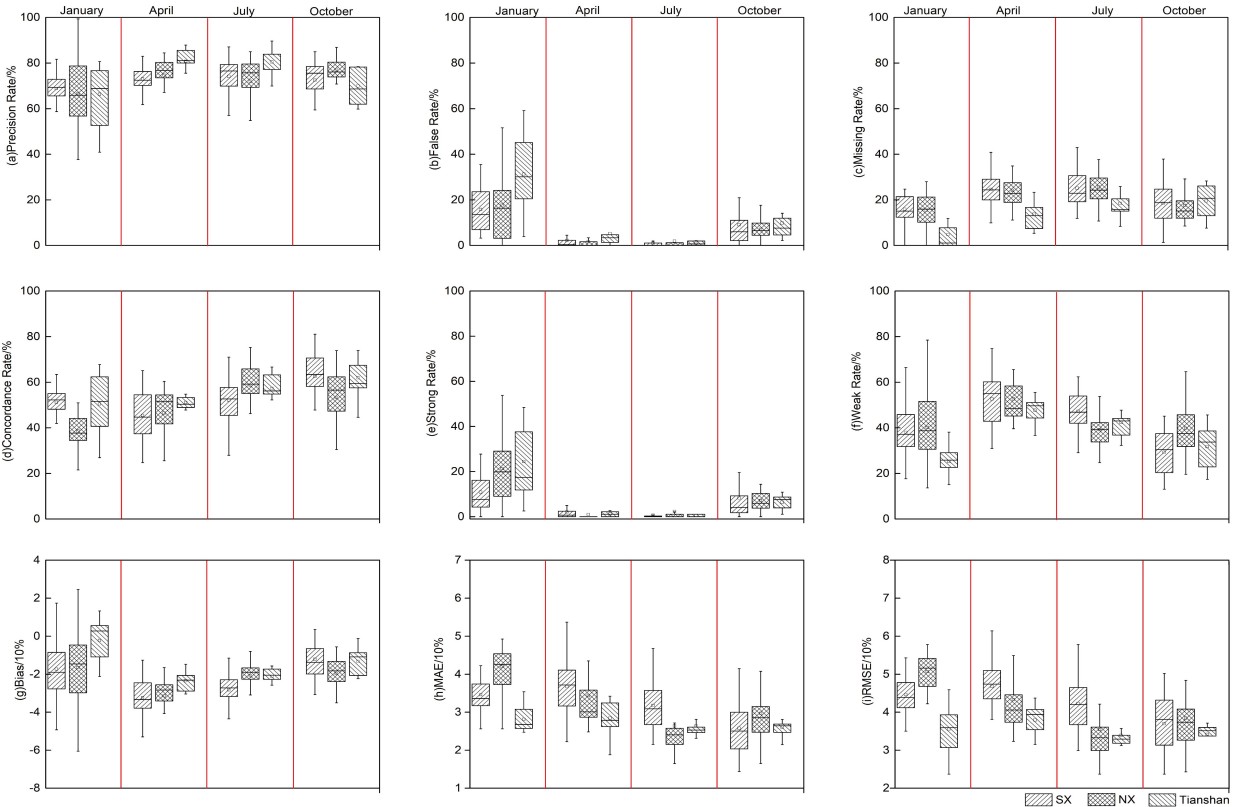

**Figure 6.** The precision, consistency and error box plot of FY-2F/CTA products in different temperature conditions of Xinjiang.

### 3.1.3 Under different altitude conditions

The altitude affects the LST, for every 100m increase in altitude, the LST would drop by approximately 0.52°C. When the

temperature difference between cloud top and underlying surface in clear sky is very small, it is difficult to distinguish

between the cloud pixels and the non-cloud pixels, which could affect the cloud recognition. The altitude of Xinjiang is

divided into four levels: less than 1000 m (31 GOS are distributed), 1000-1500 m (22 GOS are distributed), 1500-2000 m (8

GOS are distributed) and greater than 2000 m (5 GOS are distributed) in this paper. Figure 7 shows the precision,

consistency and error of FY-2F/CTA products at different altitude of Xinjiang. It is observed that with the increase of altitude,

the PR and CR of FY-2F/CTA present a slightly decreasing trend ( $k_{PR}$ , the slop of first-order linear regression of PR and

altitude, is -3.95; $R^2_{PR}$ , the coefficient of determination of PR and altitude, is 0.932 (except for the altitude of greater than

2000 m). $k_{CR}$ is -2.12; $R^2_{CR}$ is 0.544 ), but the PR increases significantly when the altitude is greater than 2000 m. The

reason for this may be mainly due to the fact that in mountainous areas with higher altitude, the cloud cover is high, and the

cloud pattern mainly consists of altostratus or nimbostratus, cirrostratus, and the cloud layer is thicker, so it is more accurate to identify the cloud and non-cloud, but less effective in the discerning of the amount of TCC.

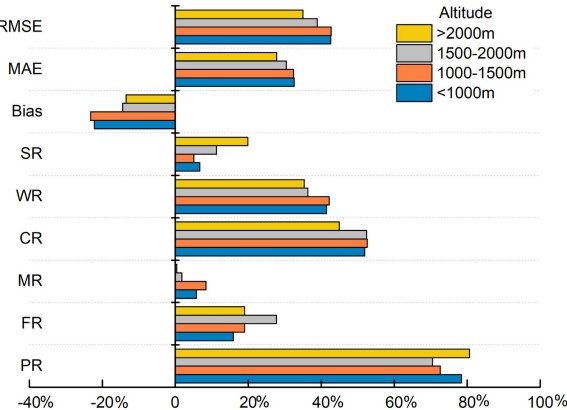

**Figure 7.** The precision, consistency and error of FY-2F/CTA products at different altitudes conditions of Xinjiang. Among them, the number of samples is 37939 for altitude less than 1000 meters, 27080 for altitude between 1000 to 1500 meters, 11232 for altitudes between 1500 to 2000 meters and 4604 for altitudes greater than 2000 meters.

## 3.2 The difference between FY-2F/CTA products and Manual observed TCC in dust and non-dust effect period of Xinjiang

As the seasons change, the types of underlying surface also change in most areas of Xinjiang, from vegetation in spring and summer, to bare land in autumn and snow in winter. Studies have shown that dusty weather occurs frequently in the Taklimakan Desert (Figure 9b), mainly in spring and summer, accounting for 88.3 % of the total number of dusty events, with winter being the season with the fewest dusty weather occurs throughout the year, accounting for only 2.3 % of the total (Zhou et al., 2017). In addition, the EOS/MODIS satellite remote sensing snow cover thematic map of Xinjiang in January (Figure 9a) shows that there is no stable snow cover in the Taklimakan desert in January, and the underlying surface changes little in the four seasons. In order to reduce the influence of underlying surface changes, only the influence of dust on cloud identification is considered, therefore, the precision, consistency and error of FY-2F/CTA are analysed by selecting January as the non-dust period and April as the dust period in Tazhong and Qiemo (Figure 8). It shows that dust can reduce the CR of FY-2F/CTA discrimination by about 15 %, but it has little effect on the PR, and the MR and WR in dust period are higher than those in non-dust period, and the FR and SR in dust period are lower than those in non-dust period. This might be caused by the fact that dust period is mainly dominated by low cloud, and the detection ability of satellites to low cloud is worse than that of high cloud, so the MR and WR during dust period are much higher than those during non-dust period. And beyond that dust storm often accompanied by precipitation, so the dusty weather has little effect on the identification of cloud and non-cloud, however, dusty weather has a greater effect on the recognition of cloudage due to uncertain sky conditions and poor visibility.

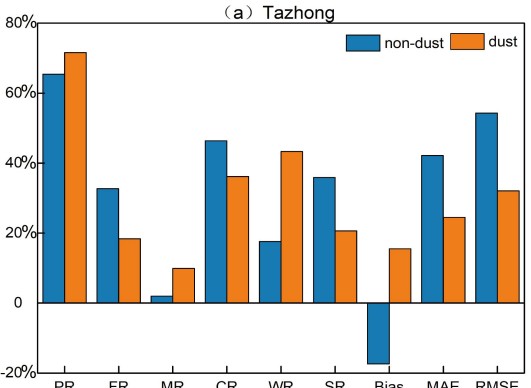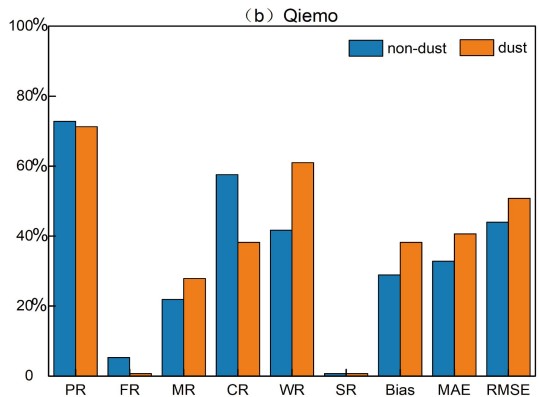

**Figure 8.** The precision, consistency and error box plot of FY-2F/CTA products in dust and non-dust effect period of Xinjiang.

In this case, the number of samples is 153 in Tazhong and 151 in Qiemo.

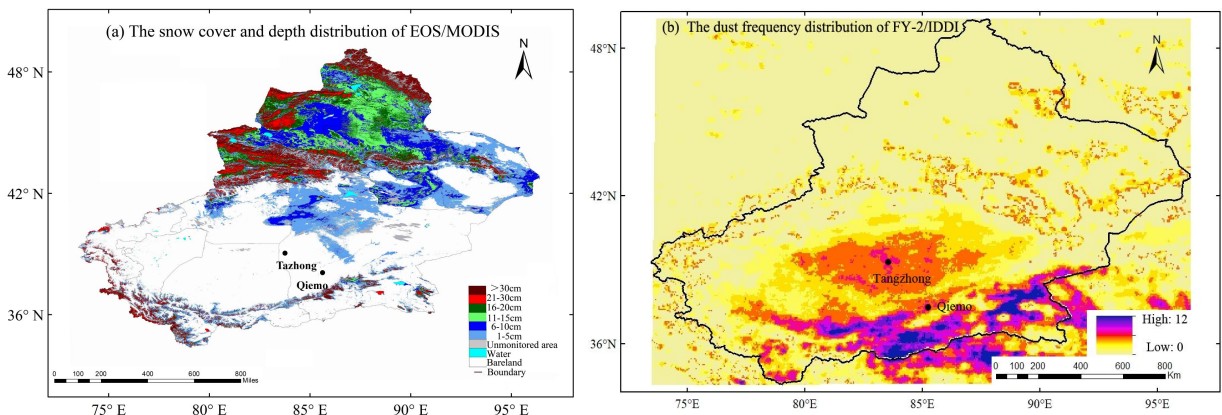

**Figure 9.** Distribution of snow cover (in January 2016) and dust frequency (in April 2016) in Xinjiang.

**(a)** The snow cover and depth distribution of EOS/MODIS, **(b)** The right picture is the dust frequency distribution of FY-2 infrared

difference dust index (FY-2/IDDI).

### 3.3 The difference between FY-2F/CTA products and Manual observed TCC under various cloud cover levels of Xinjiang

Referring to the definition of cloud cover levels in weather forecast and the classification of cloud cover level by Han et al.

(2015), taking the ground-based manual TCC observation as the standard, the amount of TCC ranges from 0 - 10 % is

defined as clear sky, 20 % - 30 % as partly cloudy, 40 % - 70 % as cloudy and 80 % - 100 % as overcast. Figure 10 shows the

precision, consistency and error of FY-2F/CTA under different cloud cover levels of Xinjiang. It can be seen that, in general,

the PR of FY-2/CTA in Xinjiang region is directly proportional to the amount of TCC, with 97.2 %, 92.0 %, 85.9 % and

31.8 % for overcast > cloudy > partly cloudy > clear sky respectively, that is, the higher the amount of TCC, the better the

PR of FY-2F/CTA identification overall. Under different cloud cover levels, the MR is high and FR is low (except clear sky).

The FR decreases with the increase of the amount of TCC, and it is the highest in clear sky, up to 64.9 %, it is zero in all

other weather conditions. The MR is the lowest in partly cloudy and the highest in cloudy, that is, partly cloudy > cloudy >

clear sky > overcast, which are 14.1 %, 8.0 %, 3.3 % and 2.9 % respectively. The CR is inversely proportional to the amount of TCC, that is, partly cloudy> clear sky > overcast > cloudy, the CR are 82.2 %, 76.6 %, 34.1 % and 30.3 % respectively, that is to say, the higher the amount of TCC, the lower the CR of FY-2F/CTA identification overall. Under different cloud cover levels, WR is high and SR is low. The SR decreases with the increase of the amount of TCC, that means, clear sky > partly cloudy >cloudy > overcast, the SR are 23.4 %, 11.0 %, 3.9 % and 0.4 % respectively. On the contrary, the WR increases with the increase of the amount of TCC, that is, cloudy day > overcast > partly cloudy > clear sky, the WR are 65.8 %, 65.5 %, 6.7 % and 0.01 % respectively. The EIs are also proportional to the amount of TCC. Under clear sky condition, the precision of FY-2/CTA increases from the central to the north and south, that is, the PR and CR are the highest, and the EIs are the lowest in SX. Under partly cloudy, cloudy, overcast conditions, the precision of FY-2/CTA decreases from the central to the south, that is, the PR and CR are the highest, and the EIs is the lowest in Tianshan. In Xinjiang region, the CR of FY-2F/CTA under clear sky and partly cloudy conditions are higher than the average value of 63 % in china (Li et al., 2018), and of about 60 % in North China and Huanghuai region (Han et al., 2015).

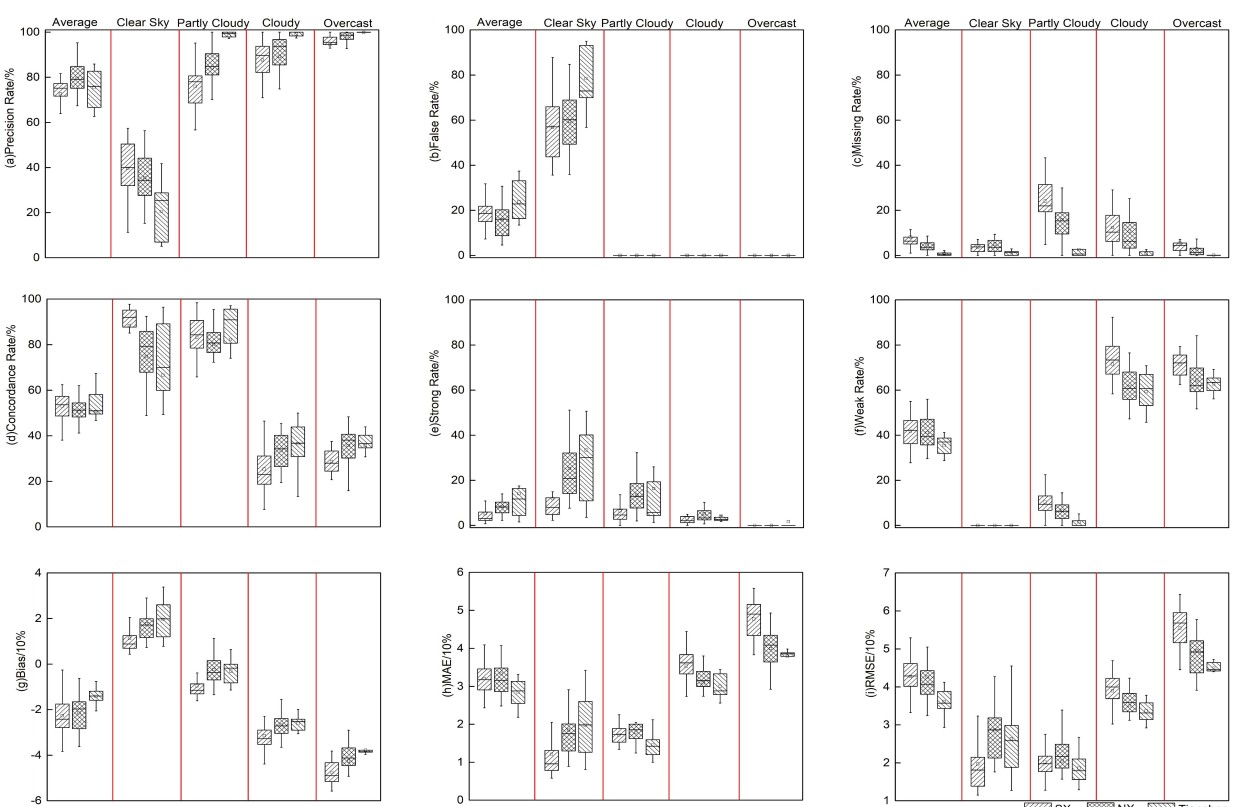

**Figure 10.** The precision, consistency and error comparison box plot of FY-2F/CTA under different TCC levels in Xinjiang. The number of samples is 24931 for clear sky, 7954 for partly cloudy, 9557 for cloudy, and 38413 for overcast.

## 4 Conclusions

Using satellite data to retrieve TCC has compensated for the limitations of traditional ground-based manual TCC

observations, and provided valuable foundational data for many studies. However, Errors are unavoidable in the process of deriving the TCC from radiometric values received by satellite, it is limited by the accuracy of many algorithms such as calibration, positioning and cloud detection, as well as the effects of underlying surface type, surface temperature and weather conditions. In this paper, Xinjiang was taken as the test area, nine evaluation indexes (PR, FR, MR, CR, SR, WR, Bias, MAE and RMSE) of FY-2F/CTA are calculated and analysed under complex underlying surface type, various temperature and altitude, dust effects and different cloud cover levels, and the precision, consistency and EIs of FY-2F/CTA are tested and evaluated, and the following main conclusions are reached.

(1) The PR of FY-2F/CTA in Xinjiang region shows a trend of gradually decreasing from north to south, and it demonstrats a high FR (the highest in Tianshan) and a low MR (the highest in SX). There is little difference in the CR among three sub-regions of Xinjiang, showing a high SR and a low WR, which means that the amount of TCC inverted from FY-2 satellite data are generally lower than that observed by GOS. The EIs increase from central to the north and south of Xinjiang, that means the EIs are the lowest in Tianshan, and the highest in SX.

(2) The FY-2F/CTA exhibit higher PR and CR in the vegetation underlying surface compared to non-vegetation. That is, FY-2F/CTA have the best identification effect on the forest and plowland underlying surface, and the worst effect on the snow and ice underlying surface. The MR in the vegetation underlying surface is higher (grassland is the highest), and the FR in the non-vegetation underlying surface is higher (ice and snow is the highest). The WR is higher than the SR in all kinds of surface underlying, but there is little difference between them in snow and ice underlying surface.

(3) With the increases of temperature, the PR and CR of FY-2F/CTA increase, while the EIs decrease. Under various temperature conditions, FY-2F/CTA have always been exhibiting high MR, low FR (on the contrary in January), high WR and low SR.

(4) With the increase of altitude, the AR and CR of FY-2F/CTA decrease, but the PR increases significantly when the altitude is higher than 2000 m.

(5) Dust reduces the CR of FY-2F/CTA, increases their WR and MR, but has a relatively minor impact on the identification of cloud and non-cloud.

(6) Under different cloud cover levels, the PR and EIs of FY-2F/CTA are proportional to the amount of TCC, while the CR is inversely proportional to it. It presents a high MR and a low FR (except clear sky), and a high SR and a low WR. That means, under clear sky conditions, the PR and CR are the highest, and the EIs are the lowest in the SX; under partly cloudy, cloudy and overcast conditions, the PR and CR are the highest, and the EIs are the lowest in Tianshan.

Although the FY-2F/CTA dataset released by the national satellite center has some systematic errors with the ground-based manual TCC observations, it should be appropriate corrected by considering the complex underlying surface conditions, the influence of dust and different cloud cover levels, which could provide better data guarantee for the research. And the data sequence length, accuracy and spatial-temporal resolution can meet the needs of most climate research we need.

### Appendix A: Abbreviations

| | |
|---|---|
| TCC | Total cloud cover |
| TCCPs | Total cloud cover products |
| FY-2F | FengYun-2F |
| FY-2F/CTA | Cloud total amount of FengYun-2F stationary satellite |
| PR | Precision rate |
| FR | False rate |
| MR | Missing rate |
| CR | Consistency rate |
| WR | Weak rate |
| SR | Strong rate |
| EIs | Error indexes |
| MAE | Mean absolute error |
| RMSE | Root mean square error |
| NX | Northern Xinjiang |
| Tianshan | Tianshan Mountains |
| SX | Southern Xinjiang |
| GOS | ground observation stations |

*Data availability.* Data published in this main paper's figures and tables are available via the figshare data repository (https://doi.org/10.6084/m9.figshare.22015592, Comparing FY-2F/CTA products to ground-based manual total cloud cover observations in Xinjiang under complex underlying surfaces and different Weather Conditions, Dataset, 2023). Underlying research data are also available by request to Li, S. (rainlishuai@163.com).

*Author contributions.* Li, S., Zhang, H., Chen, Y. H., Wang, Z. L., Li, X. Y., Li, Y., and Xue, Y. Y. designed the study. Li, Y., and Xue, Y. Y. carried out the data collection. Li, S., Wang, Z. L. and Li, X. Y. carried out the data processing and analysis. Li, S., Zhang, H., and Chen, Y. H., assisted with the interpretation of results. All co-authors contributed to writing and reviewing the paper.

*Competing interests.* The contact author has declared that none of the authors has any competing interests.

ther geographical representation in this paper. While Copernicus Publications makes every effort to include appropriate place names, the final responsibility lies with the authors.

*Acknowledgements.* We are thankful to the National Satellite Meteorological Center for FY-2F/CTA data support. We further acknowledge the researcher Zhen Z. J. and Yang C. J. of the National Satellite Meteorological Center for giving a lot of guidance in FY-2/CTA data processing. We are grateful to the engineer Cui Y. of Urumqi Meteorological Satellite Ground Station for giving good advice on the preparation of snow cover distribution in Xinjiang. We thank the editors and three anonymous reviewers, whose comments and suggestions improved the utility and readability of this paper.

*Financial support.* This work was financially supported by the National Key R&D Program of China (grant no. 2022YFC3701202), the National Natural Science Foundation of China (grant no. 42275039), and the S&T Development Fund of Chinese Academy of Meteorological Sciences (2022KJ019).

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
