# Peer review of "Comparing FY-2F/CTA products to ground-based manual total cloud cover observations in Xinjiang under complex underlying surfaces and different Weather Conditions"

_Atmospheric Measurement Techniques, 2023_

## Referee Comment (RC1)

Cloud are the important parameter in weather and climate research . In xinjiang area there are sparsed ground station and the satellite data application is more important .So the validation of satellite data is crucial. The research is meaningful. But there are some comments as following:

1. Line 120 : "2.2 Research data", the description is too simple. The distribution of the 66 ground based TCC observation should be given, such as see(figure 1). The ground based TCC is about 0 to 10? and the FY-2F/CTA should be 0-100%?

2. Line 133:"where the projection method is Mercator projection, the nearest neighbor method is used for resampling. Secondly, selecting ground-based observations that match the timing of satellite observations".
(1) Which satellite/CTA data you use? You did Mercator projection? Why use Mercator projection?
(2) "Nearest neighbor method", what distance?
(3) "match the timing" ,how many minute you use? the same time? Please give the details.

3. Line144 to Line 147:"When the observation 145 of ground station is clear sky, but the satellite detection result is cloud, then the effective cloud arithmetic average is performed on the points in a certain area around the point, and if it is still clear sky, then the satellite is judged to have missed the detection, this point is recorded as Yn." Please check is it right? It's not consistent with " ③ When the observation of ground station is clear sky but the satellite detection result is cloud, it would be judged that the satellite misjudgment and be recorded as Ny;"

4. Line 155:"they are considered to be stronger that the values of difference are greater than 2, they are considered to be weaker that the values of difference are less than -2;" and the consistency rate (CR), strong rate (SR) and weak rate (WR) can be expressed as Eq. (5) to Eq. (7) respectively.
    is there any cite paper?

5. It should give some description for the figures in the paper .For example , with Figure 2, it should tell the reader which is PR distribution, which is the MR distribution at first. The reader will not be confused and search the information in Figures.
    The following Figures have the same problems.
6. Line 179: should be 1.375um ,not "gm" .Same Line 180

7. Line 186: Line 188: Line 198:  FT-2F/CTA is not right.

8. Line 276, "It is observed that with the increase of altitude, the PR and CR of FY-2F/CTA present a decreasing trend", It seems that decreasing trend is not obvious.

9. All results should be given the numbers of sample. how many matching data author used and got the conclusion? and how about the significance test? Otherwise ,the reader can not be convinced.

10. In "3.3 The difference between FY-2F/CTA products and Manual observed TCC under various cloud cover levels of Xinjiang ". The FY-2F/CLA product's resolution is 0.1°*0.1°(one point covers 0.1°*0.1° area), ground based TCC data is the station data (scatter data). How to consider and deal with difference of the coverage of two types of data?

11. Line 352: "highe" should be "higher"

---

## Author Response (AR1)

Dear editor and dear reviewers:

On behalf of my co-authors, we thank you very much for giving us an opportunity to revise our manuscript, we appreciate editor and reviewers very much for their positive and constructive comments and suggestions in the interactive discussion of AMT preprint.

We have studied reviewers' comments carefully and have made revision which marked in red in the paper.We have tried our best to revise our manuscript according to the comments. Please see below our replies in detail. Attached please find the revised version, which we hope reviewers would be satisfied with our answers and the revision we provided.

We would like to express our great appreciation to reviewers for comments on our paper. Looking forward to hearing from you.

Yours sincerely,
Shuai Li
College of Environmental Science and Engineering Donghua University
2999 Renmin North Road, Songjiang District, Shanghai
Mobile: (86)13999856917

Email: rainlishuai@163.com

**Reviewer 1:**

**Q1. Line 120: "2.2 Research data", the description is too simple. The distribution of the 66 ground based TCC observation should be given, such as see(figure 1). The ground based TCC is about 0 to 10? and the FY-2F/CTA should be 0-100%?**

**Response:** In Figure 1, the black flags indicate the 66 ground-based observation sites in the Xinjiang region. We added data descriptions in line 114, line 117-118, and line 274-276.

In line 114: " Black flags represent the 66 TCC ground observation stations in the Xinjiang region. "

In line117-118: "of these, 24 ground observation stations are distributed in Northern Xinjiang (NX), 10 in Tianshan Mountains (Tianshan) and 32 in Southern Xinjiang (SX)."

And in line 274-276: "The altitude of Xinjiang is divided into four levels: less than 1000 m (31 ground observation stations are distributed), 1000-1500 m (22 ground observation stations are distributed), 1500-2000 m (8 ground observation stations are distributed) and greater than 2000 m (5 ground observation stations are distributed) in this paper.

The ground based TCC is 0 to 10, and the FY-2F/CTA is 0-100%. In our data processing, we multiply the ground-based observations by 10% to match them with the satellite data.

**Q2. Line 133:"where the projection method is Mercator projection, the nearest neighbor method is used for resampling. Secondly, selecting ground-based observations that match the timing of satellite observations".**
**(1) Which satellite/CTA data you use? You did Mercator projection? Why use Mercator projection?(2) "Nearest neighbor method", what distance?(3) "match the timing" ,how many minute you use? the same time? Please give the details.**

**Response:** We used the cloud total amount of FengYun-2F stationary satellite (FY-2F/CTA).

We apologize for our carelessness. At first, we made a Mercator projection with a set of data, but because the FY-2/CTA products on the website of the National Satellite Meteorological Center used the equal latitude and longitude projection, we ended up

using the data of the equal latitude and longitude projection, and forgot to modify it in the article. At present, the changes have been made in Line118-122 of the article, which is "FY-2F/CTA customized from the data service network of National Satellite Meteorological Center (http://satellite.nsmc.org.cn/portalsite/default.aspx)from June 1, 2015 to May 30, 2016, the total number of data is 8317, with a spatial resolution of 0.1º×0.1º, the time resolution is 1h, the projection method is equal latitude and longitude projection."

We reorganized the data processing methods as detailed in lines 134-140, which is "Selecting ground-based observations that match the timing of satellite observations, and the abnormal observations are eliminated through the spatial distribution of the GOS, and the stations with continuous observation for more than 20 days are selected for the preliminary quality control of the ground observation data; The satellite cloud cover data are extracted according to the time, longitude and latitude information of the GOS, and the hourly data of TCC observed by the GOS are matched with those observed by the satellite, and the total number of matched data is 80,855."

**Q3. Line144 to Line 147:"When the observation of ground station is clear sky, but the satellite detection result is cloud, then the effective cloud arithmetic average is performed on the points in a certain area around the point, and if it is still clear sky, then the satellite is judged to have missed the detection, this point is recorded as Yn." Please check is it right? It's not consistent with " ③ When the observation of ground station is clear sky but the satellite detection result is cloud, it would be judged that the satellite misjudgment and be recorded as Ny."**

**Response:** We are very grateful to the reviewer for your care and diligence in finding mistakes of the article. We have change this part into "When the observation of ground station is cloud, but the satellite detection result is clear sky, then the effective cloud arithmetic average is performed on the points in a certain area around the point, and if it is still resulted clear sky, then the satellite is judged to have missed the detection, this point is recorded as Yn". The specific modification are in Line 145-148 of the revised article.

**Q4. Line 155:"they are considered to be stronger that the values of difference are greater than 2, they are considered to be weaker that the values of difference are less than -2;" and the consistency rate (CR), strong rate (SR) and weak rate (WR) can be expressed as Eq. (5) to Eq. (7) respectively.is there any cite paper?**

**Response:** We have supplemented the relevant citations in Line 154-157, the specific content is "For the consistency analysis of FY-2F/CTA, if the absolute values of the difference between FY-2F/CTA and ground-based manual TCC observations are less than or equal to 2, they are considered to be correct; if the values of difference are greater than 2 or less than -2, they are considered to be stronger or weaker respectively (Li et al., 2018); Then the consistency rate (CR), strong rate (SR) and weak rate (WR) can be expressed as Eq. (5) to Eq. (7) respectively (Han et al., 2015)"

**Q5. It should give some description for the figures in the paper .For example, with Figure 2, it should tell the reader which is PR distribution, which is the MR distribution at first. The reader will not be confused and search the information in Figures.**

**Response:** We modified Figure 2 by enlarging all the points and legends in the figure to make the image clearer, and added the markings (a), (b)......(i) to each figure. We added the description for the figures in Line 213-215, the specific content is "Figure 2. The precision, consistency and error spatial distribution map of FY-2F/CTA products in Xinjiang. Where, from Figures (a) to (i) denote PR, FR, MR, CR, SR, WR, Bias, MAE, RMSE respectively. The total number of all valid matches is 80855, among them, 29750 are distributed in NX, 10884 are distributed in Tianshan and 40221 are distributed in SX."

[Figure]

Figure 2. The precision, consistency and error spatial distribution map of FY-2F/CTA products in Xinjiang.

Where, from Figures (a) to (i) denote PR, FR, MR, CR, SR, WR, Bias, MAE, RMSE respectively. The total

number of all valid matches is 80855, among them, 29750 are distributed in NX, 10884 are distributed in Tianshan

and 40221 are distributed in SX.

**Q6. Line 179: should be 1.375um ,not "gm" .Same Line 180.**

**Response:** We apologize for our carelessness. Here the unit is "μm", we have
modified this content in Line 180 and Line 181.

**Q7. Line 186: Line 188: Line 198: FT-2F/CTA is not right.**

**Response:** We apologize for our carelessness. Here is "FY-2F/CTA", we have
modified this content in Line 186, Line187 and Line 201.

**Q8. Line 276, "It is observed that with the increase of altitude, the PR and CR of FY-2F/CTA present a decreasing trend", It seems that decreasing trend is not obvious.**

**Response:** We agree with the comment. We have changed this sentence into " It is observed that with the increase of altitude, the PR and CR of FY-2F/CTA present a slightly decreasing trend ( $k_{PR}$ , the slop of first-order linear regression of PR and altitude, is -3.95; $R^2_{PR}$ , the coefficient of determination of PR and altitude, is 0.932 (except for the altitude of greater than 2000 m). $k_{CR}$ is -2.12; $R^2_{CR}$ is 0.544 ), but the PR increases significantly when the altitude is greater than 2000 m." in Line 277- 280.

**Q9. All results should be given the numbers of sample. how many matching data author used and got the conclusion? and how about the significance test? Otherwise ,the reader can not be convinced.**

**Response:** Thanks for your great advice. We have added a note about sample size and matching data to the description of each figure.

In line 213-215: Figure 2. The precision, consistency and error spatial distribution map of FY-2F/CTA products in Xinjiang. Where, from Figures (a) to (i) denote PR, FR, MR, CR, SR, WR, Bias, MAE, RMSE respectively. The total number of all valid matches is 80855, among them, 29750 are distributed in NX, 10884 are distributed in Tianshan and 40221 are distributed in SX.

In line 239-241: Figure 4. The precision, consistency and error of FY-2F/CTA products in complicated underlying surface of Xinjiang. In this case, the number of samples is 9196, of which 1650 are distributed in snow and ice underlying, 1596 in desert underlying, 992 in city underlying, 1653 in grassland underlying, 1653 in forest underlying, 1652 in plowland underlying.

In line 254-255: Figure 5. The scatter plot of FY-2F/CTA compared to ground-based manual TCC observations in Xinjiang. The total number of all valid matches is 264, among them, 66 in January, 66 in April, 66 in July, and 66 in October.

In line 285-287: Figure 7. The precision, consistency and error of FY-2F/CTA products at different altitudes conditions of Xinjiang. Among them, the number of samples is 37939 for altitude less than 1000 meters, 27080 for altitude between 1000

to 1500 meters, 11232 for altitudes between 1500 to 2000 meters and 4604 for altitudes greater than 2000 meters.

In line 308-309: Figure 8. The precision, consistency and error box plot of FY-2F/CTA products in dust and non-dust effect period of Xinjiang. In this case, the number of samples is 153 in Tazhong and 151 in Qiemo.

In line 338-339: Figure 10. The precision, consistency and error comparison box plot of FY-2F/CTA under different TCC levels in Xinjiang. The number of samples is 24931 for clear sky, 7954 for partly cloudy, 9557 for cloudy, and 38413 for overcast.

In line 248-250: The correlation between the two is the best in July and October, and is the worst in January, and all of them pass the significance test of 0.01 except for January.

**Q10. In "3.3 The difference between FY-2F/CTA products and Manual observed TCC under various cloud cover levels of Xinjiang ". The FY-2F/CLA product's resolution is 0.1º*0.1º(one point covers 0.1º*0.1º area), ground based TCC data is the station data (scatter data). How to consider and deal with difference of the coverage of two types of data?**

**Response:** In the data processing process, we based on the latitude and longitude of the ground observation stations, which are matched with the satellite products, and the values of the satellite products are extracted directly. Firstly, considering that clouds are different from the land surface, the spatial variation in a small area is not as drastic as that of the land surface; secondly, the ground-based artificial observation is a positive zenith observation, which may be slightly coarser than the resolution of the FY-2F/CTA products of 0.1º*0.1º. So, relatively speaking, such a direct match is feasible.

**Q11. Line 352: "highe" should be "higher".**

**Response:** Here is "higher", we have modified this content in Line 357.

We apologize for the language problems in the original manuscript. We have double checked and modified the whole word spelling.

**Q1. Lines (57-91): A comprehensive comparison between the different studies could benefit this introduction. Is there a patter of what is usually over/underestimated?**

**Response:** In general, there are two ways to calculate the total cloud cover, one is introduced in section 2.3 of this paper, the total cloud cover calculation method, mentioned in lines 126-131. And we also added specific method description in lines 131-134: ″This calculation method starts from the radiative transfer equation, takes into account the emissivity of the cloud, preserves the spatial resolution of the original observation image, and improves the subimage problem in principle, but the TCC computed by satellite will appear to be low when compared with the TCC observed by GOS″.

Another method of calculating total cloud cover is to divide pixels into clear sky pixels and complete cloud cover pixels after cloud detection. For the pixel matrix, the total cloud cover can be expressed as $TCC = N_{cld} / N$ (where, the number of total pixel in the pixel matrix is N, and the number of complete cloud pixel is $N_{cld}$). The advantage of this method is that it is simple to calculate and similar to the ground-based manual total cloud cover. However, the observed pixel matrix is considered in the calculation, and the subpixel cloud cover is ignored, which may cause overestimation of the cloud cover and decrease the spatial resolution of the data.

**Q2. Lines 92-106: The region of interest is already being introduced in 2.1. and you are giving here more detail about the data, than in the actual section. At this point it would be better to outline the paper rather than introducing things in detail.**

**Response:** Thanks for your great advice. We have reorganized this section on the presentation of FY-2 satellite data into section 2.2 Research data. In this part, the significance and outline of the paper are described. The specific modification are in Line 93-101 of the revised article: Xinjiang is a typical arid and semi-arid region, and the shortage of water resources has become a "bottleneck" problem in the development of social economy and ecological construction in Xinjiang. It is a vast and sparsely populated area with extremely low spatial coverage rate of ground-based conventional observation stations, which is more suitable for satellite observation.

How to use satellite observations to complement ground-based observation has become an urgent issue. The accuracy assessment of cloud cover retrieved by satellite is the basis of application and also a challenging task. In this paper, the examination and evaluation of the cloud total amount of FengYun-2F stationary satellite (FY-2F/CTA) are carried out using ground-based manually observed TCC and considering complex underlying surface (subsurface types, temperature and altitude conditions) and different weather conditions (dust effects and different cloud cover levels) with Xinjiang as the examination region, with a view to providing valuable references for the application and research of FY-2 cloud products.

**Q3. Figure 1: text slipped out of image caption.**

**Response:** We apologize for our carelessness. We have change this part into "Overview of the geographic location and topography of Xinjiang with three mountain ranges (Altai Mountains, Tianshan Mountains and Kunlun Mountains) and two basins (Tarim Basin and Junggar Basin). The red dashed polygons indicate the three sub-regions (NX, SX, and Tianshan), Black flags represent the 66 TCC ground observation stations in the Xinjiang region". The specific modifications are in Line 112-114 of the revised article.

**Q4. Section 2.2: Link should probably be a citation. How is the data collected? What is the satellite instrument? Radiometer?**

**Response:** The FY-2F/CTA products were downloaded from the National Satellite Weather Center Data Service (http://satellite.nsmc.org.cn/portalsite/default.aspx).

Among them, FengYun-2F (FY-2F) is the fourth geostationary satellite developed by China independently, which can provide observation data every half an hour, enabling better monitoring of the whole process of cloud formation, development and extinction. This part is introduced in lines 122-124.

**Q5. Abbreviations Ac, TCC and CTA: It is unclear whether Ac is derived for satellite or observations since TCC is used to describe the observations and not satellite. So make sure, that your variables are easy to distinct for someone not familiar with your work.**

**Response:** Thanks for your great advice. TCC is the abbreviation of Total Cloud Cover. In the process of quoting the formula, the original formula used $A_C$ to represent the total cloud cover, now we have been unified in the article, modified to TCC to represent the total cloud cover.

The specific changes were made in line 129-130:

$$TCC = (I - I_{clr})/(I - I_{cld}) \qquad (1)$$

Where TCC is the total cloud cover.

FY-2F/CTA stands for the abbreviation of cloud total amount of FengYun-2F stationary satellite, this is the fixed abbreviation for the FY-2 Total Cloud Cover product on the data website of National Satellite Weather Center.

**Q6. How many pixels are taken out of Sat data? What's the area you are considering? Does the manual cloud cover extend over the same area?**

**Response:** Thanks for your great advice, another expert raised a similar question. We have added a note about sample size and matching data to the description of each figure. And the research area is Xinjiang, the manual cloud cover extend over the same area with satellite.

In line 213-215: Figure 2. The precision, consistency and error spatial distribution map of FY-2F/CTA products in Xinjiang. Where, from Figures (a) to (i) denote PR, FR, MR, CR, SR, WR, Bias, MAE, RMSE respectively. The total number of all valid matches is 80855, among them, 29750 are distributed in NX, 10884 are distributed in Tianshan and 40221 are distributed in SX.

In line 239-241: Figure 4. The precision, consistency and error of FY-2F/CTA products in complicated underlying surface of Xinjiang. In this case, the number of samples is 9196, of which 1650 are distributed in snow and ice underlying, 1596 in desert underlying, 992 in city underlying, 1653 in grassland underlying, 1653 in forest

underlying, 1652 in plowland underlying.

In line 254-255: Figure 5. The scatter plot of FY-2F/CTA compared to ground-based manual TCC observations in Xinjiang. The total number of all valid matches is 264, among them, 66 in January, 66 in April, 66 in July, and 66 in October.

In line 285-287: Figure 7. The precision, consistency and error of FY-2F/CTA products at different altitudes conditions of Xinjiang. Among them, the number of samples is 37939 for altitude less than 1000 meters, 27080 for altitude between 1000 to 1500 meters, 11232 for altitudes between 1500 to 2000 meters and 4604 for altitudes greater than 2000 meters.

In line 308-309: Figure 8. The precision, consistency and error box plot of FY-2F/CTA products in dust and non-dust effect period of Xinjiang. In this case, the number of samples is 153 in Tazhong and 151 in Qiemo.

In line 338-339: Figure 10. The precision, consistency and error comparison box plot of FY-2F/CTA under different TCC levels in Xinjiang. The number of samples is 24931 for clear sky, 7954 for partly cloudy, 9557 for cloudy, and 38413 for overcast.

In line 248-250: The correlation between the two is the best in July and October, and is the worst in January, and all of them pass the significance test of 0.01 except for January.

**Q7. Figure 2: Fonts too small, cannot see any results.**

**Response:** We modified Figure 2 by enlarging all the points and legends in the figure to make the image clearer, and added the markings (a), (b)......(i) to each figure. We added the description for the figures in Line 213-215.

[Figure]

Figure 2. The precision, consistency and error spatial distribution map of FY-2F/CTA products in Xinjiang.

Where, from Figures (a) to (i) denote PR, FR, MR, CR, SR, WR, Bias, MAE, RMSE respectively. The total

number of all valid matches is 80855, among them, 29750 are distributed in NX, 10884 are distributed in Tianshan

and 40221 are distributed in SX.

**Q8. Figure 3: Green does not translate well visually. Maybe dark green.**

**Response:** We have adjusted the green line in Figure 3 to dark green. Details are on
Line 216.

[Figure]

**Figure 3. Taylor diagrams between FY-2F/CTA products and ground-based manual TCC observations**

**in different regions of Xinjiang.**

**Q9. Line 175: What is Figure 2 actually showing?**

**Response:** We added the description for the figures in Line 213-215, the specific content is "Figure 2. The precision, consistency and error spatial distribution map of FY-2F/CTA products in Xinjiang. Where, from Figures (a) to (i) denote PR, FR, MR, CR, SR, WR, Bias, MAE, RMSE respectively. The total number of all valid matches is 80855, among them, 29750 are distributed in NX, 10884 are distributed in Tianshan and 40221 are distributed in SX."

**Q10. Line 195: Why is the same as NOAA/AVHRR TCCP? Where are the results for NOAA presented?**

**Response:** We added the reason in Line 197-200: This may be due to the fact that NOAA/AVHRR TCCPs and FY-2/CAT use the same calculation method, and that the resolution and channel information of the two satellites are close. Meanwhile, the differences in observational capabilities and cloud detection algorithms between MODIS/Aqua TCCPs and the two satellite TCCPs mentioned above are the main reasons for the deviations (Liu et al., 2016, 2017).
This result for NOAA is cited in the article of Liu et al. (Liu et al, 2016, 2017).
The specific article is:

Liu, J., Yang, X. F., and Cui, P.: Validation of total cloud amount in 2007 derived by NOAA/AVHRR, Plateau Meteor. (in Chinese), 35(4), 1027–1038, https://doi.org/10.7522/j.issn.1000-0534.2015.00029, 2016.

Liu, J., Cui, P., and Xiao, M.: The bias analysis of FY-2G cloud fraction in summer and winter, J. Appl. Meteor. Sci. (in Chinese), 28(2), 177–188, https://doi.org/10.11898/1001-7313.20170205, 2017.

**Q11. Figure 5-10: All too small.**

**Response:** The resolution of all the graphs in this article has been adjusted to 600dpi. The PDF file required by the editorial department will reduce the resolution of the article images, but we uploaded the higher resolution images (600dbi) as an attachment1.

**Q12. Most of the figures are too small and the paper still contains many typos and spelling mistakes (capitol E for Earth for example)**

**Response:** We apologize for the language problems in the original manuscript. We have double checked and modified the whole word spelling.

---

## Author Response (AR2)

Dear editor and reviewer:

On behalf of my co-authors, we thank you very much for giving us an opportunity to revise our manuscript, we appreciate editor and reviewers very much for their positive and constructive comments and suggestions, especially in the part of research data and methods.

We have studied reviewer's comments carefully and have made revision which marked in red in the paper. We have tried our best to revise our manuscript according to the comments. Please see below our replies in detail. Attached please find the revised version, which we hope reviewer would be satisfied with our answers and the revision we provided.

We would like to express our great appreciation to reviewer for comments on our paper. Looking forward to hearing from you.

Yours sincerely,
Shuai Li
College of Environmental Science and Engineering Donghua University
2999 Renmin North Road, Songjiang District, Shanghai
Mobile: (86)13999856917

Email: rainlishuai@163.com

**Reviewer 3:**

**Q1. Line 15: You say the ground-based observations are manually observed TCC. However, in the answer to reviewer 1, Q10, you write that the ground-based observation are artificial positive zenith observations? Can you clarify within the paper what kind of ground-based observations were used.**

**Response:** We apologize for the ambiguity caused by our earlier answer to reviewer 1, Q10. The ground-based observations are manually observed TCC, that means, human-eye observation on the ground. We have clarified that in Line 118-119: Manually observed TCC are the human-eye observations on the ground.

**Q2. Further, the matching principle is not clear: manually observed TCC are according to the WMO synoptic code in oktas and satellite observation are given in percentage (0 to 100). If it is manually observed TCC. What is the unit with which TCC is observed and what are average visibility ranges at the station? How do you convert the ground-based observation to 0-100%? All this information should be provided to the reader in the paper. If it is artificial positive zenith observation, what instrument is used? What is the error of the instrument and what is the measurement principle?**

**Response:** The ground-based observations are manually observed TCC. We added the specifications of manually observed TCC and the TCC conversion method in Line 118-127 and 148-160.

In Line 118-127: Manually observed TCC are the human-eye observations on the ground, they are collected five times a day (at 00:00, 03:00, 06:00, 09:00, and 12:00 UTC), with values ranging from 0 to 10. The observations follow the specifications outlined below: when the sky is entirely clear, the TCC is recorded as 0; if the sky is completely covered by clouds, it is recorded as 10; if the sky is fully covered by clouds but openings in the clouds allow glimpses of the sky, it is recorded as 10-; When there are a few clouds in the sky, amounting to less than 0.5 of the sky's coverage, the TCC is recorded as 0; When visibility is impaired due to phenomena such as haze, suspended dust, sandstorms, or blowing sand, rendering the determination of TCC either entirely or partially indiscernible, the TCC is recorded as " – ". If clouds occupy one-tenth of the sky, the TCC is recorded as 1; if they occupy

two-tenths of the sky, it is recorded as 2, and so forth, following a similar progression for different levels of cloud coverage.

In Line 148-160: The specific data processing methods are as follow. ① To mitigate the impact of short-term weather changes on ground observations, reduce data fluctuations caused by observational errors, and enhance data stability, the stations with continuous observations for 20 days or more are selected to enhance data stability (Liu et al., 2016); the abnormal observations, including missing data and outliers (observations < 0 or > 10 or the records are " – " ), are removed from the dataset during the preliminary quality control of the ground observation data. ② The ground observation TCC reflects the cloud cover within a certain range around each observation point, the area can reach several kilometers or even more than ten kilometers, for satellite observation TCCPs, only the radiation ratio at grid points is considered. Therefor for satellite products, the TCC at each station is determined by averaging the cloud amounts of all grid points within a 10 km radius centered on the station's location. ③ Using the observation time, latitude and longitude information of the observation stations, the TCC observed by the GOS are matched with those observed by the satellite, and the total number of matched data is 80,855.

In addition, because FY-2F/CTA observations are provided as integer values from 0 % to 100 %, they are converted into tenths from 0 to 10, as listed in Table 1 (Kim et al., 2023).

**Table 1. Tenth cloud cover conversion table of satellite (%) and ceilometer (okta) cloud cover.**

| % | ≤5 | 5-15 | 15-25 | 25-35 | 35-45 | 45-55 | 55-65 | 65-75 | 75-85 | 85-95 | >95 |
|------|----|------|-------|-------|-------|-------|-------|-------|-------|-------|---------|
| Okta | 0 | 1 | 2 | 2 | 3 | 4 | 5 | 6 | 6 | 7 | 8 |
| Tenth | 0 | 1 | 2 | 3 | 4 | 5 | 6 | 7 | 8 | 9 | 10- / 10 |

**Q3. Line 136: What do you mean with abnormal observations? Please clarify what criteria you have used to filter the ground-based observations.**

**Response:** We added the description of abnormal observations in Line 150-152: the abnormal observations, including missing data and outliers (observations < 0 or > 10 or the records are " – "), are removed from the dataset during the preliminary quality control of the ground observation data.

**Q4. Line 137: Why do you only use data from stations with more than 20 days of continuous observations? Please provide more information.**

**Response:** The reason we used data from stations with more than 20 days of continuous observations is: To mitigate the impact of short-term weather changes on ground observations, reduce data fluctuations caused by observational errors, and enhance data stability, the stations with continuous observations for 20 days or more are selected to enhance data stability (Liu et al., 2016). We added this part in Line 148-150.

**Q5. Line 138: The matching criteria of ground and satellite data are not sufficiently described in the paper. Please provide the information on the used radius of the FY-2F/CTA observations around the ground-based station. Also as asked by reviewer 2, Q6, the number of pixels taken out of the satellite should be provide in the method part of the paper. Response to question Q6 from reviewer 2: How can the manual cloud cover extend be the same as the satellite? The manual TCC depends on the visibility at time of observation at the station. Please state clearly in the method part what the radius around the station is you have used for the comparison of the satellite observations. The sample size and matching radius should be explained in the methods part. The same holds for the allowed time difference you used for the comparison.**

**Response:** We added the matching criteria of ground and satellite data in Line 152-156: The ground observation TCC reflects the cloud cover within a certain range around each observation point, the area can reach several kilometers or even more than ten kilometers, for satellite observation TCCPs, only the radiation ratio at grid points is considered. Therefor for satellite products, the TCC at each station is determined by averaging the cloud amounts of all grid points within a 10 km radius centered on the station's location.

**Q6. The matching principle and the precision analysis is similar to Kim at al. (2023). The paper should be cited.**

**Response:** We have carefully read the article by Kim et al. (2023) and cited it in Line 158-159.

**Q7. Line 169: please state clear that X0 are the data from the observation site and X those from the satellite.**

**Response:** We added the description of X and $X_0$ in Line 190-191: Where N represents the number of matched samples, X are the FY-2F/CTA observations and $X_0$ are the ground-based manual TCC observations.

**Q8. The number of matches given in the Figure captions are not clear. According to Figure 2, you have in total of 80855 matches. But in Figure 5 the total number is 264. In the paper the figures need more explanation. What data are used, how do you filter the data, and the number of matching points.**

**Response:** We added the description of data in Line 235-238 and Line 277-282.

In Line 235-238 Figure 2. The precision, consistency and error spatial distribution map of FY-2F/CTA products in Xinjiang. Where, from Figures (a) to (i) denote PR, FR, MR, CR, SR, WR, Bias, MAE, RMSE respectively. The data is based on the hourly TCC of FY-2F/CTA and ground-based manual observations. The total number of all valid matches is 80855, among them, 29750 are distributed in NX, 10884 are distributed in Tianshan and 40221 are distributed in SX.

In Line 277-282: Figure 5. The scatter plot of FY-2F/CTA compared to ground-based manual TCC observations in Xinjiang. The data is the monthly average TCC of FY-2F/CTA and ground-based manual observations. It is based on the hourly data of FY-2F/CTA and GOS in January, April, July and October (the total sample points of 66 GOS are 7634, 7235, 7592, 7554, respectively), and after summing and average calculation, the monthly average TCC of FY-2F/CTA and ground-based manual observations of stations are obtained. Therefore, in this figure, the total number of all valid matches is 264, among them, 66 in January, 66 in April, 66 in July, and 66 in October.

**Q9. The FY-2F/CTA products (as for example written in the Figure caption 6) should be introduced in the methods part of the paper.**

**Response:** We added the introduction of FY-2F/CTA products in Line 130-137:

Among them, FengYun-2F (FY-2F) is the fourth geostationary satellite developed by China independently. It is equipped with various detection channels, including visible light (0.5 - 0.9 μm), mid-wave infrared (3.5 - 4.0 μm), thermal infrared (infrared channel 1(10.3 - 11.3 μm), infrared channel 2(11.5 - 12.5 μm) and water vapor (6.3 - 7.6 μm)). The satellite provides observational data every half hour, allowing for improved monitoring of the entire process of cloud formation, development, and dissipation. The cloud products of FY-2F include cloud cover, cloud type, cloud top temperature, among others. FY-2F/CTA represents its TCC product, the spatial resolution is 0.1º × 0.1º, temporal resolution is 1 hour, and the projection method is equal latitude and longitude projection. This configuration enables enhanced monitoring capabilities for the complete lifecycle of clouds.

---

## Author Response (AR3)

Dear editor :

On behalf of my co-authors, we thank you very much for providing us with the opportunity to publish our manuscript in AMT, we appreciate editor and reviewers for their positive and constructive comments and suggestions.

We have prepared a short video abstract, and uploaded it to the AMT website supplement. We have uploaded the short video abstract to the TIB AV-Portal, but it has not yet been approved, so we have not received the DOI address. Once it passes the review, I will send you the DOI address via email. We are unsure whether it meets the requirements. Should you require any further modifications or adjustments, please do not hesitate to contact me.

Once again, thank you for your support and encouragement. I look forward to the publication of the manuscript in AMT.

Looking forward to hearing from you.

Yours sincerely,

Shuai Li

College of Environmental Science and Engineering, Donghua University

2999 Renmin North Road, Songjiang District, Shanghai

Mobile: (86)13999856917

Email: rainlishuai@163.com